# Effect of Behavioral Weight Management Interventions Using Lifestyle mHealth Self-Monitoring on Weight Loss: A Systematic Review and Meta-Analysis

**DOI:** 10.3390/nu12071977

**Published:** 2020-07-03

**Authors:** Iván Cavero-Redondo, Vicente Martinez-Vizcaino, Rubén Fernandez-Rodriguez, Alicia Saz-Lara, Carlos Pascual-Morena, Celia Álvarez-Bueno

**Affiliations:** 1Health and Social Research Center, Universidad de Castilla-La Mancha, 16071 Cuenca, Spain; ivan.cavero@uclm.es (I.C.-R.); ruben.fernandez12@alu.uclm.es (R.F.-R.); Alicia.delsaz@uclm.es (A.S.-L.); carlos.pascual@uclm.es (C.P.-M.); Celia.alvarezbueno@uclm.es (C.Á.-B.); 2Universidad Politécnica y Artística del Paraguay, Asunción 001518, Paraguay; 3Facultad de Ciencias de la Salud, Universidad Autoónoma de Chile, Talca 3460000, Chile; 4Movi-Fitness S.L, Universidad de Castilla La-Mancha, 16002 Cuenca, Spain

**Keywords:** obesity, mHealth, self-monitoring

## Abstract

Alongside an increase in obesity, society is experiencing the development of substantial technological advances. Interventions that are easily scalable, such as lifestyle (including diet and physical activity) mobile health (mHealth) self-monitoring, may be highly valuable in the prevention and treatment of excess weight. Thus, the aims of this systematic review and meta-analysis were to estimate the following: (i) the effect of behavioral weight management interventions using lifestyle mHealth self-monitoring on weight loss and (ii) the adherence to behavioral weight management interventions using lifestyle mHealth self-monitoring. MEDLINE via PubMed, EMBASE, the Cochrane Central Register of Controlled Trials and the Web of Science databases were systematically searched. The DerSimonian and Laird method was used to estimate the effect of and adherence to behavioral weight management interventions using lifestyle mHealth self-monitoring on weight loss. Twenty studies were included in the systematic review and meta-analysis, yielding a moderate decrease in weight and higher adherence to intervention of behavioral weight management interventions using lifestyle mHealth self-monitoring, which was greater than other interventions. Subgroup analyses showed that smartphones were the most effective mHealth approach to achieve weight management and the effect of behavioral weight management interventions using lifestyle mHealth self-monitoring was more pronounced when compared to usual care and in the short-term (less than six months). Furthermore, behavioral weight management interventions using lifestyle mHealth self-monitoring showed a higher adherence than: (i) recording on paper at any time and (ii) any other intervention at six and twelve months.

## 1. Introduction

Overweight and obesity are the fifth highest risk factors for global death, which corresponds to about 3.4 million deaths yearly, making them a global public health priority and a health challenge [1]. According to the World Health Organization (WHO), the obesity rate has tripled since the 1980s. In 2014, 39% of the adult population was classified as overweight and 13% as obese [2,3], with an estimated one billion people classified as overweight and 573 million people as obese [4] in 2030 if there is no attenuation of the current weight gain trends.

The increase in the prevalence of overweight and obesity and their association with many chronic diseases, namely cardiovascular diseases, type 2 diabetes and some cancers, has sparked the interest of researchers and scientific institutions looking for effective ways to promote a healthy lifestyle and weight control [1]. Among behavioral weight loss approaches, lifestyle (including diet and physical activity) and behavior strategies (including self-monitoring) have been consistently related to short- and long-term weight loss management [5] The lifestyle self-monitoring approach consists of registering all food and beverages consumed, portion sizes and methods of preparation, as well as the amount of physical activity performed throughout the day, making individuals aware of their current behaviors [6]. Although, the use of self-monitoring in behavior change has a strong theoretical foundation, completing daily paper records appears to be quite tedious for most individuals [7].

Alongside the increase in obesity, society is experiencing the development of substantial technological advances. While it is true that the boom in new technologies could be involved in the increased availability of energy dense and processed foods as well as the growing prevalence of sedentary behaviors, it is also true that they could play an important role in the management of health problems [8] Interventions based on new technologies could be easily developed and accepted for the management of weight disorders, since they represent an interesting tool to increase individuals’ awareness of the quantity and quality of food consumed and the physical activity performed [9] Additionally, lifestyle mobile health (mHealth) self-monitoring [10] appears to have a greater effect on self-efficacy, patient motivation and adherence to treatment, in such a way that it may elicit a greater weight loss than conventional methods [11]. A recent systematic review concluded that mHealth applications may represent an effective strategy for weight loss, and, since the review included both observational and experimental studies, the authors called for an update, including a meta-analysis, when the number of intervention studies allowed for the pooled effect size (ES) of the effect of mHealth applications on weight loss to be estimated [12].

At the present time, considering that obesity prevalence rates do not appear to be decreasing, interventions that are easily scalable, such as lifestyle (diet and physical activity) mHealth self-monitoring, are increasingly valuable in the prevention and treatment of excess weight. However, because no previous meta-analysis has synthesized the effect of lifestyle mHealth self-monitoring as part of behavioral weight management approach, the aims of this systematic review and meta-analysis were to estimate the following: (i) the effect of behavioral weight management interventions using lifestyle mHealth self-monitoring on weight loss and (ii) the adherence to behavioral weight management interventions when lifestyle mHealth self-monitoring was used.

## 2. Materials and Methods

Before conducting this systematic review and meta-analysis, we registered it in the PROSPERO database (registration number ID: CRD42020164608). We followed the Cochrane Handbook for Systematic Reviews of Interventions [13] to conduct it and the Preferred Reporting Items for Systematic Reviews and Meta-Analyses (PRISMA) [14] to report it.

### 2.1. Search Strategy

MEDLINE (via PubMed), EMBASE, the Cochrane Central Register of Controlled Trials and the Web of Science databases were systematically searched, from their inception until March 2020. We searched for experimental studies comparing the effects of behavioral weight management interventions using lifestyle (diet and physical activity) mHealth self-monitoring on weight loss. The search strategy for the MEDLINE database is displayed in Appendix A
Table A1. To complete the systematic literature search, we examined the references of the eligible articles.

### 2.2. Study Selection

The included studies had to meet the following inclusion criteria: (i) participants—general population; (ii) design—randomized controlled trials (RCTs), non-randomized controlled trials (non-RCTs) and pilot studies; (iii) type of interventions—studies comparing the effect of lifestyle (diet and physical activity) mHealth self-monitoring (i.e., personal digital assistants (PDAs), smartphones or web-based); and (iv) outcomes—weight change and adherence to behavioral weight management interventions using lifestyle mHealth self-monitoring. The criteria for the exclusion of studies were as follows: (i) non-eligible publication types, such as review articles, editorials, comments, guidelines or case-reports; and (ii) duplicate reports—when this was the case, we extracted the data from the different reports and included in this systematic review the one providing the most detailed data.

### 2.3. Data Extraction and Risk of Bias Assessment

An ad-hoc table summarized the following information from the original reports: (1) year of publication; (2) country; (3) study design; (4) sample characteristics (sample size, age distribution and type of population); (5) baseline means of adiposity parameters (weight, body mass index (BMI) and waist circumference (WC)); (6) type of intervention (PDA, smartphone or web-based); (7) comparison groups; (8) length of intervention; and (9) percentage of dropouts.

The Cochrane Collaboration’s tool for assessing the risk of bias (RoB2) [15] was used to assess the risk of bias of the included RCTs. The evaluation of six domains is included in this tool: randomization process, deviations from intended interventions, missing outcome data, measurement of the outcome and selection of the reported result. Each domain could be assessed as having a low risk of bias, some concerns or a high risk of bias.

For non-RCTs, the ROBINS-I tool was used [16]. This tool evaluates the risk of bias according to seven domains: bias due to confounding, bias in the selection of participants for the study, bias in the measurement of interventions, bias due to deviations from intended interventions, bias due to missing data, bias in the measurement of outcomes and bias in the selection of the reported result. Overall bias could be considered as “low risk of bias” if all domains were classified as “low risk”, “moderate risk of bias” if all domains were classified as “low risk” or “moderate risk”, “serious risk of bias” if there was at least one domain rated as “serious risk”, “critical risk of bias” if there was at least one domain rated as “critical risk” and “no information” if there was no clear indication that the study had a serious or critical risk of bias and there was a lack of information in one or more domains.

The literature search, data extraction and quality assessment were conducted by two independent reviewers (IC-R and RF-R), and a third reviewer (CA-B) was included when inconsistencies remained after discussion. Kappa statistics was calculated to assess the agreement rate between reviewers.

### 2.4. Statistical Analysis and Data Synthesis

To compute the pooled estimate of the ES and its 95% confidence intervals (CIs) for weight change, we used the DerSimonian and Laird method [17]. A standardized mean difference score was calculated, using Cohen’s d index as the ES statistic, in which negative ES values indicate a weight loss in favor of behavioral weight management interventions using lifestyle mHealth self-monitoring. Cohen’s d values represented the following: (i) weak effects when values were around 0.2, (ii) moderate effects when values were around 0.5, (iii) strong effects when values were around 0.8 and (iv) very strong effects when values were greater than 1.0 [18]. Additionally, a pooled estimate of the mean weight change difference in kg was calculated.

Adherence to behavioral weight management interventions using lifestyle mHealth self-monitoring was calculated as the risk of dropping out of the lifestyle mHealth self-monitoring group versus other interventions or the control group. Relative risk (RR) was used as the risk estimate.

The heterogeneity of the results across studies was assessed using the I^2^ statistic. I^2^ values were interpreted as: might not be important (0–40%); may represent moderate heterogeneity (30–60%); substantial heterogeneity (50–90%); or considerable heterogeneity (75–100%). The corresponding *p*-values were also considered.

Subgroup analyses were performed based on the type of mHealth intervention (PDA, smartphone or web-based), the type of comparison group (usual care, paper record or wait-list) and the length of the intervention (≤3 months, six months and ≥12 months). Sensitivity analyses were conducted to assess the robustness of the summary estimates and to detect whether any particular study accounted for a large proportion of heterogeneity. Random-effects meta-regressions were used to investigate whether the results were associated with the age of participants and the baseline means of weight, BMI or WC, since these variables may explain the observed heterogeneity.

Finally, the Egger test [19] (*p* < 0.10 considered as statistically significant [20]) and a visual inspection of the funnel plots were used to assess publication bias. STATA SE software, version 15 (StataCorp, College Station, TX, USA), was used for the statistical analyses.

## 3. Results

### 3.1. Systematic Review

Twenty studies [21,22,23,24,25,26,27,28,29,30,31,32,33,34,35,36,37,38,39,40] (Figure 1) addressing the effect of behavioral weight management interventions using lifestyle mHealth self-monitoring on weight loss were identified, which were conducted in six countries: 12 in the United States [21,23,25,28,31,33,34,35,37,38,39,40], two in the United Kingdom [24,32], three in Australia [22,26,29], one in New Zealand [27], one in South Korea [30] and one in Finland [36]. Reports were published between 2007 and 2019, and they included studies using the following experimental designs: 17 RCTs [21,22,23,24,25,26,27,28,29,31,32,33,34,35,36,37,38] and three non-RCTs [30,39,40]. Regarding the characteristics of the included populations, participants were aged between 20.5 and 59.8 years, with sample sizes ranging from 11 to 131 participants in the lifestyle mHealth self-monitoring intervention groups and from six to 133 participants in the control groups. The baseline weight, BMI and WC of the studies ranged from 62.1 kg to 116.9 kg, from 27.0 kg/m^2^ to 40.1 kg/m^2^ and from 88.2 cm to 120.4 cm, respectively (Table 1).

The mHealth interventions were delivered through PDAs, smartphones and web-based approaches, while comparator groups included usual care, paper records and wait-lists. The length of the interventions ranged from one to 24 months. Eight studies performed analyses for more than one time point [24,27,28,33,35,36,37,38]. Three studies included two intervention arms [24,31,39]. Additionally, 16 studies specified the application used for lifestyle mHealth self-monitoring: Lose It! [21,34,38,39], TXT2BFiT [22], Dietmate Pro [23], My Meal Mate [24], MyFitnessPal [25,27,37], Be Positive Be Healthy [26], SmartLoss [27], CalorieKing [28,34,39] and Fitbit [30]. The percentage of dropouts from the lifestyle mHealth self-monitoring intervention group ranged from 5.0% to 54.8% (Table 2).

### 3.2. Risk of Bias

For RCTs, as evaluated by the RoB2 tool, 47.4% of studies showed some concerns regarding the risk of bias and 52.6% showed a high risk for overall bias (mainly as a consequence of a high risk of bias in the measurement of the outcome domain) (see Appendix A
Figure A1). Among non-RCTs, as evaluated by the ROBINS-I tool, the risk of bias was scored as moderate in 33.3% of studies and serious in 66.7% (mainly as a consequence of a serious risk of bias in the missing data domain) (see Appendix A
Figure A2).

### 3.3. Meta-Analysis

The pooled ES of behavioral weight management interventions using lifestyle mHealth self-monitoring on weight loss was −0.37 (95%CI: −0.54, −0.19). Additionally, the pooled mean difference in weight was −1.78 kg (95%CI: −2.70, −0.85). The heterogeneity between studies was substantial (I2 = 84.6%; *p* < 0.001) (Figure 2). The pooled RR for dropping out of the lifestyle mHealth self-monitoring group was 0.78 (95%CI: 0.63, 0.96). The heterogeneity between studies was moderate (I2 = 37.8%; *p* = 0.049) (Figure 3).

### 3.4. Subgroup Analyses

Subgroup analyses considering the type of mHealth intervention and the type of comparison group showed that a greater effect was observed when the mHealth intervention used a smartphone (ES = −0.36; 95%CI: −0.51, −0.13, I2 = 56.4%) and usual care was the control group (ES = −0.51; 95%CI: −0.83, −0.20, I2 = 84.0%). Additionally, based on the length of the intervention, a pooled ES for a weight loss of −1.08 (95%CI: −1.55, −0.62, I2 = 87.6%) was estimated for ≤3 months and of −0.23 (95%CI: −0.49, −0.02, I2 = 70.4%) for six months. Regarding adherence, there were less dropouts in mHealth interventions when paper records were used for the control group (RR = 0.63; 95%CI: 0.44, 0.91, I2 = 20.9%). Additionally, there were less dropouts from the mHealth interventions in studies performed for six months (RR = 0.76; 95%CI: 0.59, 0.97, I2 = 46.0%) and for twelve months (RR = 0.79; 95%CI: 0.64, 0.96, I2 = 0.0%) (Table 3).

### 3.5. Sensitivity Analyses

The pooled ES estimate was not significantly different when data from each individual study were removed from the analyses one at a time.

### 3.6. Meta-Regressions

The random-effects meta-regression models for the effects on weight loss and adherence showed that age (*p* = 0.365 and 0.462) and baseline means of weight (*p* = 0.724 and 0.593), BMI (*p* = 0.440 and 0.979) and WC (*p* = 0.677 and 0.428) were not related to heterogeneity across studies (see Appendix A
Table A2).

### 3.7. Publication Bias

Evidence of publication bias was found in both the funnel plot asymmetry and Egger’s test (*p* = 0.016 for effect on weight loss and *p* = 0.008 for adherence).

## 4. Discussion

This systematic review and meta-analysis provides an overview of the evidence supporting lifestyle (diet and physical activity) mHealth self-monitoring, as part of a behavioral weight management approach, as a suitable intervention for weight management in adults with overweight or obesity, resulting in a moderate decrease in weight and higher adherence to intervention, greater than with other interventions. Additionally, this meta-analysis shows that interventions delivered through smartphones are the most effective mHealth approach to achieve weight management in adult populations with overweight or obesity. The effect of behavioral weight management interventions using lifestyle mHealth self-monitoring interventions was more pronounced when they were compared to usual care and in the short-term (less than six months). Furthermore, behavioral weight management interventions using lifestyle mHealth self-monitoring interventions showed a higher adherence than: (i) paper records at any time and (ii) any other intervention at six months and twelve months.

A previous systematic review on the effect of mHealth applications on weight loss highlighted that the use of these interventions is widely accepted, easy to use and helpful in achieving weight loss goals [12]. Additionally, there is evidence for the consistent and significant positive relationship between lifestyle (diet and physical activity) and behavior (self-monitoring) strategies and successful weight management [5]. Likewise, our systematic review and meta-analysis supports the notion that mHealth self-monitoring interventions have a moderate effect on reducing weight, which may represent a mean weight loss of 1.78 kg greater than with other intervention types.

Our results not only show a positive effect on weight loss but also fewer dropouts of subjects included in mHealth interventions in the short and long-term. The mechanisms through which behavioral weight management interventions using lifestyle mHealth self-monitoring are effective may be explained from two perspectives: the user and the clinical setting [41]. For users, mHealth interventions enhance patients’ self-efficacy and empowerment and improve daily life autonomy and adherence to treatment [42]. Moreover, the mHealth approach reduces contact with the clinical setting and, as a consequence, decreases the workload for health care workers (physicians, nurses and nutritionists, especially in primary care) [43,44].

There is a variety of devices that could be used for mHealth. Our subgroup analyses support the notion that smartphones, a technology available to a high proportion of the population worldwide [45], are the most effective mHealth devices for weight management. This, along with the high prevalence rates of both physical inactivity and obesity, has triggered a growing interest in the development of smartphone applications for health, fitness and diet, which have increased exponentially in the last few years [46]. More than half of smartphone users may have downloaded a health application [47]; however, the use of these applications for clinical outcomes is still very limited and even non-existent in most contexts [48].

The limitations of this study are as follows: (1) the risk of bias assessment showed that a few studies presented some concerns or moderate risk of bias, while most showed a high or serious risk of bias. It should be noted that the main reason behind the high risk of bias in the included studies was the impracticality of blinding interventions, but this limitation is difficult to overcome in this type of intervention. (2) The lack of studies using devices that did not allow for comparison between them, with reliable results only being obtained for smartphones. (3) The intervention groups could be very different considering that they used different types of applications; however, they have the common characteristic that they were carried out through an mHealth device. (4) Many of the studies did not control for the effect of other covariates which could affect the results, such as educational or socioeconomic level. (5) Regarding the analysis of adherence, it cannot be assumed that lifestyle mHealth self-monitoring is the only reason for participants to dropout, since these interventions are usually part of a broader behavioral weight management approach. (6) Finally, this meta-analysis showed publication bias, mainly due to the lack of studies with small sample sizes.

Even with the risk of being branded as opportunists, we are not reluctant to emphasize the importance of mHealth interventions in times when face-to-face contact must be limited, such as those we are living in currently, particularly in clinical settings, in which the transmission of infectious diseases, namely Influenza or Sars-CoV-2, may be greater. Additionally, everything appears to indicate that the current concept of treatment will lead to an increase in the use of mHealth in daily clinical practice [49].

## 5. Conclusions

In summary, our study demonstrates that lifestyle mHealth self-monitoring interventions, as part of a behavioral weight management approach, are suitable interventions for short-term weight management in adults with overweight/obesity. Considering our results and the population’s accessibility to smartphones, this type of device could be a useful and largely scalable tool for weight management. Thus, future well designed RCTs and controlled clinical trials with higher statistical power are essential in order to reinforce the evidence, which is still weak, to demonstrate that effective mHealth interventions could eventually change the current paradigm of lifestyle prescription, increasing patients’ self-management of disease, developing new clinical practice guidelines and facilitating workflow in everyday clinical consultations.

## Figures and Tables

**Figure 1 nutrients-12-01977-f001:**
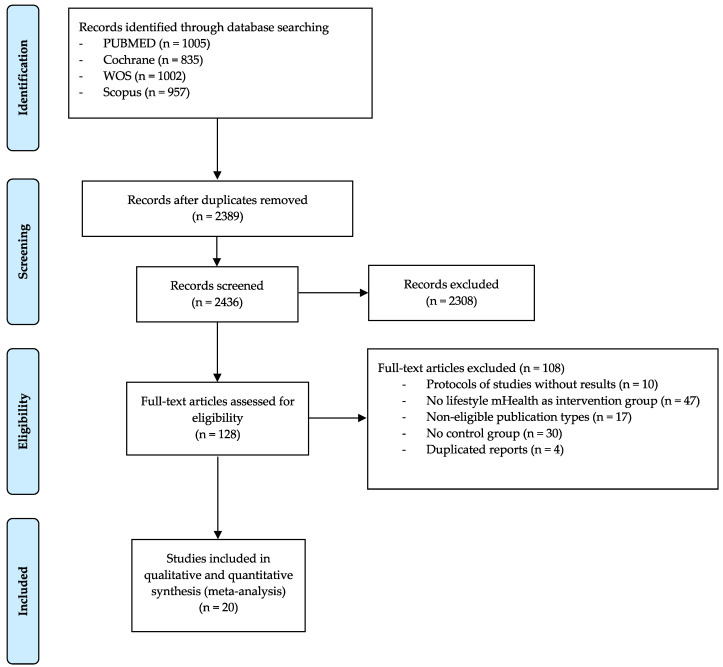
Preferred Reporting Items for Systematic Reviews flowchart.

**Figure 2 nutrients-12-01977-f002:**
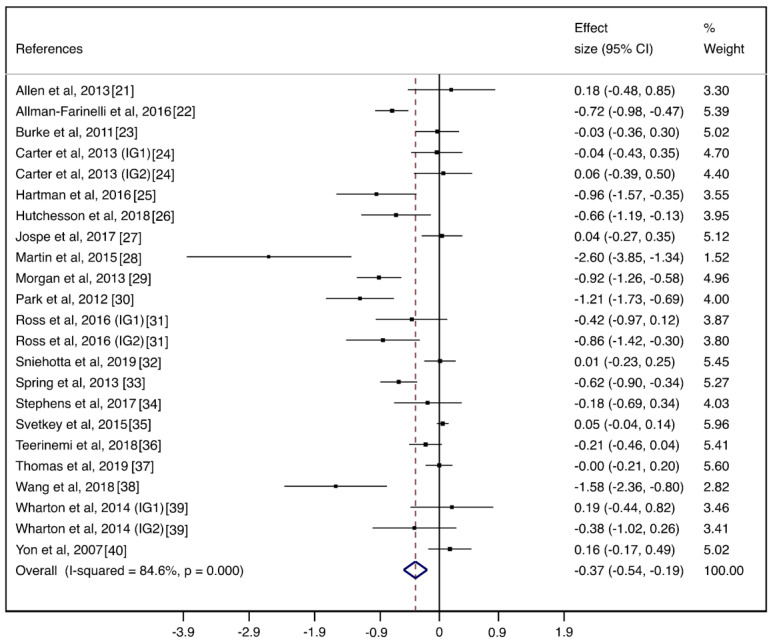
Forest plots of the pooled effect size for behavioral weight management interventions using lifestyle mHealth self-monitoring on weight loss.

**Figure 3 nutrients-12-01977-f003:**
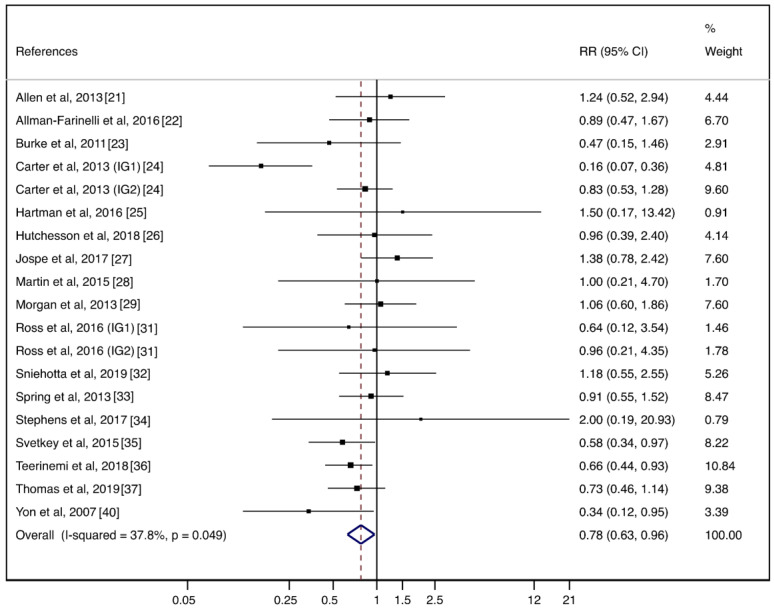
Forest plots of the pooled relative risk for the adherence of behavioral weight management interventions using lifestyle mHealth self-monitoring.

**Table 1 nutrients-12-01977-t001:** Characteristics of studies included in the meta-analysis.

Reference	Country	Study Design	Mean Age (Years)	Sample Size	Baseline Weight (kg Mean ± SD)	Baseline BMI (kg/m^2^ Mean ± SD)	Baseline WC (cm Mean ± SD)
Allen et al., 2013 [21]	USA	RCT	CG: 42.5 ± 12.1 IG: 45.3 ± 13.2	CG: 18 IG: 17	CG: 96.0 ± 17.4 IG: 96.4 ± 16.9	CG: 34.1 ± 4.1 IG: 35.3 ± 4.1	CG: 112.4 ± 11.5 IG: 109.7 ± 17.1
Allman-Farinelli et al., 2016 [22]	Australia	RCT	CG: 27.2 ± 4.9 IG: 28.1 ± 4.9	CG: 125 IG: 123	CG: 79.3 ± 12.6 IG: 78.4 ± 11.2	CG: 27.0 ± 2.7 IG: 27.3 ± 2.3	NR
Burke et al., 2011 [23]	USA	RCT	CG: 47.4 ± 8.5 IG: 46.7 ± 9.2	CG: 72 IG: 68	NR	CG: 33.9 ± 4.6 IG: 33.5 ± 3.8	CG: 109.5 ± 11.6 IG: 111.3 ± 11.1
Carter et al., 2013 [24]	UK	RCT	CG: 42.5 ± 8.3 IG1: 41.2 ± 8.5 IG2: 41.9 ± 10.6	CG: 19 IG1: 40 IG2: 20	CG: 97.9 ± 18.7 IG1: 96.4 ± 16.0 IG2: 96.4 ± 19.9	CG: 34.5 ± 5.7 IG1: 33.7 ± 4.2 IG2: 34.5 ± 5.6	NR
Hartman et al., 2016 [25]	USA	RCT	CG: 59.8 ± 5.9 IG: 59.4 ± 5.6	CG: 17 IG: 33	CG: 85.3 ± 10.5 IG: 86.3 ± 10.2	CG: 31.3 ± 3.7 IG: 32.2 ± 3.4	NR
Hutchesson et al., 2018 [26]	Australia	RCT	CG: 27.9 ± 5 IG: 27.1 ± 4.7	CG: 28 IG: 29	CG: 79.2 ± 10.3 IG: 79.8 ± 10	CG: 29.4 ± 2.5 IG: 29.3 ± 2.5	CG: 88.2 ± 8.0 IG: 88.8 ± 9.0
Jospe et al., 2017 [27]	New Zealand	RCT	CG: 46.7 ± 11.4 IG: 44.4 ± 10.2	CG: 36 IG: 36	CG: 91.0 ± 14.9 IG: 99.1 ± 17.3	CG: 32.3 ± 4.3 IG: 33.2 ± 4.8	CG: 99.8 ± 11.0 IG: 102.7 ± 12.8
Martin et al., 2015 [28]	USA	RCT	CG: 43.3 ± 2.6 IG: 45.6 ± 2.7	CG: 20 IG: 20	CG: 80.6 ± 2.9 IG: 80.0 ± 2.3	CG: 29.5 ± 3.2 IG: 30.2 ± 2.7	CG: 94.5 ± 2.1 IG: 93.2 ± 2.2
Morgan et al., 2013 [29]	Australia	RCT	CG1: 48.0 ± 11.2 CG2: 48.0 ± 10.8 IG: 46.5 ± 11.1	CG1: 52 CG2: 54 IG: 53	CG1: 103.8 ± 15.0 CG2: 101.8 ± 12.4 IG: 104.7 ± 14.5	CG1: 33.1 ± 3.9 CG2: 32.4 ± 3.3 IG: 32.8 ± 3.4	CG1: 113.6 ± 9.9 CG2: 112.6 ± 9.2 IG: 113.7 ± 9.7
Park et al., 2012 [30]	South Korea	Non-RCT	CG: 57.6 ± 5.5 IG: 55.8 ± 5.7	CG: 33 IG: 34	CG: 62.5 ± 9.0 IG: 62.1 ± 7.1	NR	CG: 89.6 ± 9.9 IG: 89.9 ± 5.5
Ross et al., 2016 [31]	USA	RCT	CG: 54.2 ± 9.5 IG1: 46.2 ± 13.5 IG2: 52.9 ± 10.3	CG: 26 IG1: 27 IG2: 27	CG: 91.6 ± 14.5 IG1: 89.2 ± 15.6 IG2: 87.1 ± 12.4	NR	NR
Sniehotta et al., 2019 [32]	UK	RCT	CG: 41.6 ± 11.4 IG: 42.0 ± 11.6	CG: 133 IG: 131	CG: 85.2 ± 15.7 IG: 85.1 ± 17.5	CG: 30.8 ± 5.2 IG: 30.9 ± 5.5	CG: 94.6 ± 14.7 IG: 93.6 ± 13.4
Spring et al., 2013 [33]	USA	RCT	CG: 57.7 ± 10.2 IG: 57.7 ± 13.5	CG: 35 IG: 34	CG: 110.1 ± 15.1 IG: 113.7 ± 16.1	CG: 35.8 ± 3.8 IG: 36.9 ± 5.4	CG: 120.4 ± 8.9 IG: 120.4 ± 14.0
Stephens et al., 2017 [34]	USA	RCT	CG: 20.5 ± 1.7 IG: 20.8 ± 2.0	CG: 30 IG: 29	CG: 79.6 ± 11.8 IG: 86.0 ± 16.3	CG: 29.5 ± 4.3 IG: 30.9 ± 4.4	CG: 97.0 ± 11.3 IG: 98.5 ± 10.8
Svetkey et al., 2015 [35]	USA	RCT	CG: 29.6 ± 4.3 IG: 29.2 ± 4.2	CG: 123 IG: 122	NR	CG: 35.1 ± 7.5 IG: 35.7 ± 8.2	NR
Teerinemi et al., 2018 [36]	Finland	RCT	CG:46.5 ± 10.2 IG:47.0 ± 9.4	CG: 59 IG: 70	CG: 88.6 ± 11.1 IG: 88.7 ± 10.9	CG: 30.5 ± 2.3 IG: 30.3 ± 2.0	NR
Thomas et al., 2019 [37]	USA	RCT	55.1 ± 9.9	CG: 56 IG: 114	95.9 ± 17.0	35.2 ± 5.0	NR
Wang et al., 2018 [38]	USA	RCT	CG1: 49.2 ± 10.2 CG2: 56.1 ± 5.4 IG: 58.8 ± 5.9	CG1: 6 CG2: 9 IG: 11	CG1: 92.1 ± 2.4 CG2: 116.9 ± 13.1 IG: 106.9 ± 15.1	CG1: 33.7 ± 2.7 CG2: 40.1 ± 7.0 IG: 38.9 ± 9.0	NR
Wharton et al., 2014 [39]	USA	Non-RCT	CG: 40.8 ± 3.8 IG1: 43.7 ± 3.5 IG2: 41.5 ± 4	CG: 20 IG1: 19 IG2: 18	CG: 82.2 ± 20.3 IG1: 84.2 ± 13.4 IG2: 86.1 ± 22.3	CG: 28.9 ± 1.0 IG1: 29.9 ± 0.9 IG2: 31.0 ± 1.7	NR
Yon et al., 2007 [40]	USA	Non-RCT	CG: 46.1 ± 9.2 IG: 48.2 ± 8.7	CG:93 IG: 57	CG: 86.4 ± 13.7 IG: 90.2 ± 14.0	CG: 30.9 ± 3.5 IG: 32.3 ± 3.4	NR

USA: United States of America; UK: United Kingdom; RCT: randomized control trials; CG: Control group; IG: intervention group; NR: not reported; BMI: body mass index; WC: waist circumference.

**Table 2 nutrients-12-01977-t002:** Characteristics of type of interventions in the meta-analysis.

Reference	Intervention	Comparison	Length (Months)	Dropouts (%)
Allen et al., 2013 [21]	Smartphone (Lose It!)	Usual care	6	CG: 33.3 IG: 41.2
Allman-Farinelli et al., 2016 [22]	Web-based (TXT2BFiT)	Usual care	9	CG: 14.4 IG: 12.8
Burke et al., 2011 [23]	PDA (Dietmate Pro)	Paper record	6	CG: 12.5 IG: 5.9
Carter et al., 2013 [24]	IG1: Smartphone (My Meal Mate) IG2: Web-based	Paper record	1.5 and 6	6-week follow-up: CG: 34.9 IG1: 9.3 IG2: 35.7 6-month follow-up: CG: 53.5 IG1: 7.0 IG2: 54.8
Hartman et al., 2016 [25]	Smartphone (MyFitnessPal)	Usual care	6	CG: 5.6 IG: 8.3
Hutchesson et al., 2018 [26]	Smartphone (Be Positive Be Healthy)	Wait-list	6	CG: 25.0 IG: 24.1
Jospe et al., 2017 [27]	Smartphone (MyFitnessPal)	Usual care	6 and 12	6-month follow-up: CG: 8.3 IG: 20.0 12-month follow-up: CG: 25.0 IG: 28.0
Martin et al., 2015 [28]	Smartphone (SmartLoss)	Usual care	1, 2 and 3	CG: 5.0 IG: 5.0
Morgan et al., 2013 [29]	Web-based (CalorieKing)	CG1: Wait-list CG2: Usual care	3	CG1: 7.7 CG1: 9.2 IG: 9.4
Park et al., 2012 [30]	Web-based	Wait-list	3	NR
Ross et al., 2016 [31]	IG1: Smartphone (Fitbit) IG2: Smartphone (Fitbit) + phone call	Paper record	6	CG: 11.5 IG1: 7.4 IG2: 11.1
Sniehotta et al., 2019 [32]	Web-based	Usual care	12	CG: 7.6 IG: 9.0
Spring et al., 2013 [33]	PDA	Usual care	3, 6 and 9	3-month follow-up: CG: 14.3 IG: 11.8 6-month follow-up: CG: 20.0 IG: 14.7 9-month follow-up: CG: 17.1 IG: 20.6
Stephens et al., 2017 [34]	Smartphone (LoseIt!)	Usual care	3	CG: 3.2 IG: 6.5
Svetkey et al., 2015 [35]	Smartphone (CalorieKing)	Wait-list	6, 12 and 24	6-month follow-up: CG: 15.4 IG: 5.7 12-month follow-up: CG: 13.8 IG: 10.7 24-month follow-up: CG: 14.6 IG: 14.8
Teerinemi et al., 2018 [36]	Web-based	Usual care	12 and 24	12-month follow-up: CG: 23.6 IG: 13.2 24-month follow-up: CG: 10.1 IG: 9.9
Thomas et al., 2019 [37]	Smartphone (MyFitnessPal)	Paper record	6, 12 and 18	6-month follow-up: CG: 14.3 IG: 7.9 12-month follow-up: CG: 28.6 IG: 22.8 18-month follow-up: CG: 33.9 IG: 19.3
Wang et al., 2018 [38]	Smartphone (LoseIt!)	CG1: Usual care CG2: Paper record	3 and 6	NR
Wharton et al., 2014 [39]	IG1: Smartphone (LoseIt!) IG2: Smartphone (Memo function of the smartphone)	Paper record	2	NR
Yon et al., 2007 [40]	PDA (Calorie King’s Handheld Diet Diary)	Paper record	6	CG: 19.0 IG: 7.0

CG: Control group; IG: intervention group; PDA: personal digital assistant; NR: not reported.

**Table 3 nutrients-12-01977-t003:** Subgroup analyses for pooled effect on weight loss and adherence to mHealth group according to type of mHealth intervention, type of comparison and length of intervention.

	Effect on Weight Loss				Adherence to mHealth			
Subgroup	*n*	Effect Size (95%CI)	I2	*p*	*n*	Relative Risk (95%CI)	I2	*p*
*Type of mHealth intervention*								
Smartphone	14	−0.36 (−0.58, −0.13)	56.4	0.005	11	0.76 (0.51, 1.16)	54.5	0.015
PDA	4	−0.17 (−0.46, 0.13)	62.7	0.045	3	0.61 (0.32, 1.15)	40.6	0.186
Web-based	2	0.02 (−0.21, 0.24)	0.0	0.838	5	0.83 (0.66, 1.04)	0.0	0.556
*Type of Comparison*								
Usual care	10	−0.51 (−0.83, −0.20)	84.0	<0.001	10	0.97 (0.72, 1.30)	0.0	0.677
Paper record	11	−0.22 (−0.46, 0.02)	68.4	<0.001	7	0.63 (0.44, 0.91)	20.9	0.270
Wait-list	4	−0.56 (−1.14, 0.01)	94.1	<0.001	3	0.93 (0.59, 1.48)	0.0	0.697
*Length of Intervention*								
≤3 months	14	−1.08 (−1.55, −0.62)	87.6	<0.001	7	0.91 (0.61, 1.35)	0.0	0.474
Six months	15	−0.23 (−0.44, −0.02)	70.4	<0.001	15	0.76 (0.59, 0.97)	46.0	0.035
≥12 months	5	0.02 (−0.07, 0.11)	0.0	0.432	6	0.79 (0.64, 0.96)	0.0	0.530

PDA: Personal digital assistant.

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
