# Peer review of "Effect of Behavioral Weight Management Interventions Using Lifestyle mHealth Self-Monitoring on Weight Loss: A Systematic Review and Meta-Analysis"

_nutrients, 2020, doi:10.3390/nu12071977_

Round 1
Reviewer 1 Report
This paper describes a systematic review and meta-analysis of publications meeting specified criteria for comparing mHealth interventions to control groups. Findings hold potential of contributing to the scientific literature, and it is generally well-written. This reviewer has a few minor comments, as detailed below.
- Figure 1: In the 2nd side box (“Full text articles excluded”), it is not clear what is meant by “RCT protocols”. Please clarify.
- Table 1 holds a lot of great information. What seems to be missing is information about adherence and dropouts in each study’s different groups. This is one of the paper’s aims. If it is too difficult to squeeze that into this already full table, it could be added as a small table.
- Discussion line 237: The word “increased” in this sentence may be misleading. It probably does not mean increased over the course of interventions, since adherence tends to decline over time in most studies. Do the authors intend to say it is higher compared to other approaches? If so, then specify which ones.
- Discussion, lines 240-241: (The sentence beginning with “For users” and ending with “treatment”) Please provide at least one reference for these statements.
- Discussion Line 247: What is meant by “unusual”?
Author Response
This paper describes a systematic review and meta-analysis of publications meeting specified criteria for comparing mHealth interventions to control groups. Findings hold potential of contributing to the scientific literature, and it is generally well-written. This reviewer has a few minor comments, as detailed below.
Authors
Thank you for the reviewer’s comment
Specific comment
- Figure 1: In the 2nd side box (“Full text articles excluded”), it is not clear what is meant by “RCT protocols”. Please clarify.
Authors
Thanks for the reviewer´s comment. As suggested, we have replaced ‘RCT protocols’ with ‘Protocols of studies without results’ in Figure 1.
Specific comment
- Table 1 holds a lot of great information. What seems to be missing is information about adherence and dropouts in each study’s different groups. This is one of the paper’s aims. If it is too difficult to squeeze that into this already full table, it could be added as a small table.
Authors
The reviewer’s comment seems judicious. As suggested, we have divided the information into two tables to make the information easier to read. Additionally, we have included the dropout information in the new Table 2.
Specific comment
- Discussion line 237: The word “increased” in this sentence may be misleading. It probably does not mean increased over the course of interventions, since adherence tends to decline over time in most studies. Do the authors intend to say it is higher compared to other approaches? If so, then specify which ones.
Authors
We would like to thank the reviewer’s comment. As suggested, we have modified de misleading sentence as follows:
“[…], but also fewer dropouts of subjects included in mHealth interventions in the short and long-term than ….”
Specific comment
- Discussion, lines 240-241: (The sentence beginning with “For users” and ending with “treatment”) Please provide at least one reference for these statements.
Authors
Thank you for the reviewer´s comment. As suggested, we have included the following reference to support this information:
“Farley H. Promoting self‐efficacy in patients with chronic disease beyond traditional education: A literature review. Nursing Open 2020, 7(1), 30-41.”
Specific comment
- Discussion Line 247: What is meant by “unusual”?
Authors
We would like to thank the reviewer´s comment and apologize for the misleading term. We have replaced ‘an unusual interest’ with ‘a growing interest’.

Reviewer 2 Report
This paper is a welcome addition to the literature on mHealth approaches to self-monitoring within behavioral weight control interventions in the midst of the persistent global obesity challenge.
The main point of concern that should be addressed in this manuscript is how lifestyle mHealth self-monitoring is defined and discussed. How the phrase is used implies that self-monitoring is the sole component of the studies included, which is a misstatement. Rather a phrase that indicates mHealth self-monitoring as part of behavioral weight control approach is needed and should be applied throughout the paper.
Mainly, it is concerning that self-monitoring is not discussed as one facet of multi-component behavioral interventions for weight control. While self-monitoring is a key behavior, it is not the totality of modern approaches, including those cited in this review. This can be done in the introduction and reiterated in the discussion.
The definition of adherence (aim two) is also problematic. Again, authors appear to be speaking about behavioral weight control interventions that include a mHealth self-monitoring component, but the phrase "lifestyle mHealth self-monitoring" only speaks to the self-monitoring component. Traditionally, adherence to self-monitoring implies participant adherence to using the mHealth self-monitoring tool within the behavioral intervention, and that is not what the authors mean in this paper. Thus, aim two should be edited to specifically state that adherence is actually participants remaining in the study where mHealth self-monitoring is a feature.
The discussion does amend the problematic phrasing by including the word interventions, this or a similar phrase should be consistent throughout the manuscript. Similar to the introduction, authors must discuss mHealth self-monitoring as one part of behavioral weight control interventions in the discussion. Currently, there is no mention to the full intervention here.
Limitations should also speak to the assumption of using participant drop out as a measure of mhealth self-monitoring adherence. Given that the self-monitoring tool was not the sole element of behavioral trials, it is too great a leap to assume that it is the only reason for participant drop out.
Author Response
This paper is a welcome addition to the literature on mHealth approaches to self-monitoring within behavioral weight control interventions in the midst of the persistent global obesity challenge.
Authors
We appreciate the reviewer´s comment.
Specific comment
The main point of concern that should be addressed in this manuscript is how lifestyle mHealth self-monitoring is defined and discussed. How the phrase is used implies that self-monitoring is the sole component of the studies included, which is a misstatement. Rather a phrase that indicates mHealth self-monitoring as part of behavioral weight control approach is needed and should be applied throughout the paper.
Authors
The reviewer’s comment seems judicious. As suggested, we have included the information regarding the use of mHealth self-monitoring as part of behavioral weight control approach throughout the paper.
Specific comment
Mainly, it is concerning that self-monitoring is not discussed as one facet of multi-component behavioral interventions for weight control. While self-monitoring is a key behavior, it is not the totality of modern approaches, including those cited in this review. This can be done in the introduction and reiterated in the discussion.
Authors
Thanks for the reviewer´s comment. As suggested, we have included this information in the Introduction and Discussion sections as follows:
Introduction section:
“Among behavioral weight loss approaches, lifestyle (including diet and physical activity) and behavior strategies (including self-monitoring) have been consistently related to short- and long-term weight loss management [5] The lifestyle self-monitoring approach consists of registering all food and beverages consumed, portion sizes and methods of preparation, as well as the amount of physical activity performed throughout the day, being aware individuals of their current behaviors [6]. Although, the use of self-monitoring in behavior change has a strong theoretical foundation, completing daily paper records appears to be quite tedious for most individuals [7].”
Discussion section
“Additionally, there is evidence for the consistent and significant positive relationship between lifestyle (diet and physical activity) and behavior (self-monitoring) strategies and successful weight management [5].”
Specific comment
The definition of adherence (aim two) is also problematic. Again, authors appear to be speaking about behavioral weight control interventions that include a mHealth self-monitoring component, but the phrase "lifestyle mHealth self-monitoring" only speaks to the self-monitoring component. Traditionally, adherence to self-monitoring implies participant adherence to using the mHealth self-monitoring tool within the behavioral intervention, and that is not what the authors mean in this paper. Thus, aim two should be edited to specifically state that adherence is actually participants remaining in the study where mHealth self-monitoring is a feature.
Authors
The reviewer’s comment seems judicious. As suggested, we have rephrased the aim two as follows:
“ii) the adherence to behavioral weight management interventions when lifestyle mHealth self-monitoring was used.”
Specific comment
The discussion does amend the problematic phrasing by including the word interventions, this or a similar phrase should be consistent throughout the manuscript. Similar to the introduction, authors must discuss mHealth self-monitoring as one part of behavioral weight control interventions in the discussion. Currently, there is no mention to the full intervention here.
Authors
The reviewer’s comment seems judicious. As suggested, we have replaced the term ‘lifestyle mHealth self-monitoring intervention’ with ‘behavioral weight management interventions using lifestyle mHealth self-monitoring’ throughout the paper.
Specific comment
Limitations should also speak to the assumption of using participant drop out as a measure of mhealth self-monitoring adherence. Given that the self-monitoring tool was not the sole element of behavioral trials, it is too great a leap to assume that it is the only reason for participant drop out.
Authors
Thank you for the reviewer´s comment. As suggested, we have included a statement in Limitations section as follows:
“(5) Regarding the analysis of adherence, it cannot be assumed that lifestyle mHealth self-monitoring is the only reason for participants to dropout since these interventions are usually part of a broader behavioral weight management approach.”
